# communications
## earth & environment

# Heat stress vulnerability and critical environmental limits for older adults

S. Tony Wolf [1]✉, Rachel M. Cottle[1,2], Kat G. Fisher[1], Daniel J. Vecellio [2] & W. Larry Kenney[1,2,3]

The present study examined heat stress vulnerability of apparently healthy older vs. young adults and characterized critical environmental limits for older adults in an indoor setting at rest (Rest) and during minimal activity associated with activities of daily living. Critical environmental limits are combinations of ambient temperature and humidity above which heat balance cannot be maintained (i.e., becomes uncompensable) for a given metabolic heat production. Here we exposed fifty-one young (23 ± 4 yrs) and 49 older (71 ± 6 yrs) adults to progressive heat stress across a wide range of environments in an environmental chamber during Minimal Activity (young and older subjects) and Rest (older adults only). Heat compensability curves were shifted leftward for older adults indicating age-dependent heat vulnerablity ($p < 0.01$). During Minimal Activity, critical environmental limits were lower in older compared to young adults ($p < 0.0001$) and lower than those at Rest ($p < 0.0001$). These data document heat vulnerability of apparently healthy older adults and define critical environmental limits for indoor settings in older adults at rest and during activities of daily living, and can be used to develop evidence-based recommendations to minimize the dele-terious impacts of extreme heat events in this population.

[1] Department of Kinesiology, The Pennsylvania State University, University Park, PA 16802, USA. [2] Center for Healthy Aging, The Pennsylvania State University, University Park, PA 16802, USA. [3] Graduate Program in Physiology, The Pennsylvania State University, University Park, PA 16802, USA.
✉email: stwolf@uga.edu

Anthropongenic climate change has caused Earth's average temperature to increase 1.1 °C since the latter part of the 19th century[1]. At the same time, the world's population is rapidly aging. The number of people ≥65 years old in the U.S. projected to increase to nearly 80 million by the year 2040[2]. People ≥65 years exhibit disproportionate increases in morbidity and mortality during heat waves[3]. Thus, a rapidly growing aged population coupled with an increased frequency and severity of heat waves combines to increase the number of people at risk during environmental extremes. A wealth of laboratory data confirms the mechanistic underpinnings of age-associated declines in thermoregulatory function (decrements in sweating, skin blood flow, excess cardiac strain, etc.)[4–8]. Yet the contributions of age-related physiological changes to heat vulnerability among apparently healthy older men and women remains unclear.

Additionally, although it is well-documented that adults over the age of 65 comprise the majority of excess deaths during extreme heat events, the specific combinations of ambient dry-bulb temperature ($T_{db}$) and humidity above which an age disparity begins to occur remain unclear. Critical environmental limits are defined as those combinations of environmental conditions (e.g., $T_{db}$ and humidity) above which heat balance cannot be maintained for a given metabolic heat production[9–13], and core temperature rises continuously (i.e., heat stress becomes uncompensable). Our laboratory has previously developed and refined a progressive heat stress protocol to identify critical environmental limits for prolonged indoor heat exposure across a wide range of environments, metabolic rates, and clothing ensembles[11,13–20]. Most humans who succumb to the effects of extreme heat are carrying out daily activities in and around their home, highlighting the need for establishing temperature-humidity combinations above which heat stress cannot be maintained in indoor settings[21].

Our laboratory recently published critical environmental limits for healthy young adults during low-intensity physical activity approximating the metabolic demands of activities of daily living (ADL)[15,22]. Those data serve as a best-case scenario against which to compare more vulnerable populations, including adults over the age of 65 years. Given the continued warming climate and aging population, identifying those environments above which older adults are at greatest risk of heat-related morbidity and mortality while in the home or other indoor settings is important. Empirically derived critical environmental limits, grounded in physiology and biophysics, can be used to develop evidence-based alert communication, policy decisions, triage for impending heat events, and implementation of other safety interventions for those who are most vulnerable during extreme heat events.

Therefore, the purpose of the current investigation was to (1) document increased heat vulnerability of apparently healthy older adults based on thermal balance across adverse indoor environmental conditions, and (2) establish and compare critical environmental limits for older adults during seated rest and minimal physical activity associated with ADL (MinAct). We also compared critical environmental limits for older adults during MinAct to an updated data set from that previously-published[15,22] from young adults. We hypothesized that (1) older adults would exhibit uncompensable heat stress at lower combinations of ambient $T_{db}$ and humidity than young adults, and (2) these critical environmental limits would be lower for older adults during MinAct relative to seated rest.

## Results

**Subject characteristics.** Fifty-one young (23 ± 4 yrs) and 49 older (71 ± 6 yrs) adults were randomly assigned (Fig. 1) to complete experimental trials in two environmental conditions during MinAct (young and older subjects) and seated rest (Rest; older adults only). Subjects were representative of the population in these age groups with respect to body size and aerobic fitness (Table 1)[23]. Maximal aerobic capacity ($\dot{V}O_{2max}$) was significantly lower in older adults

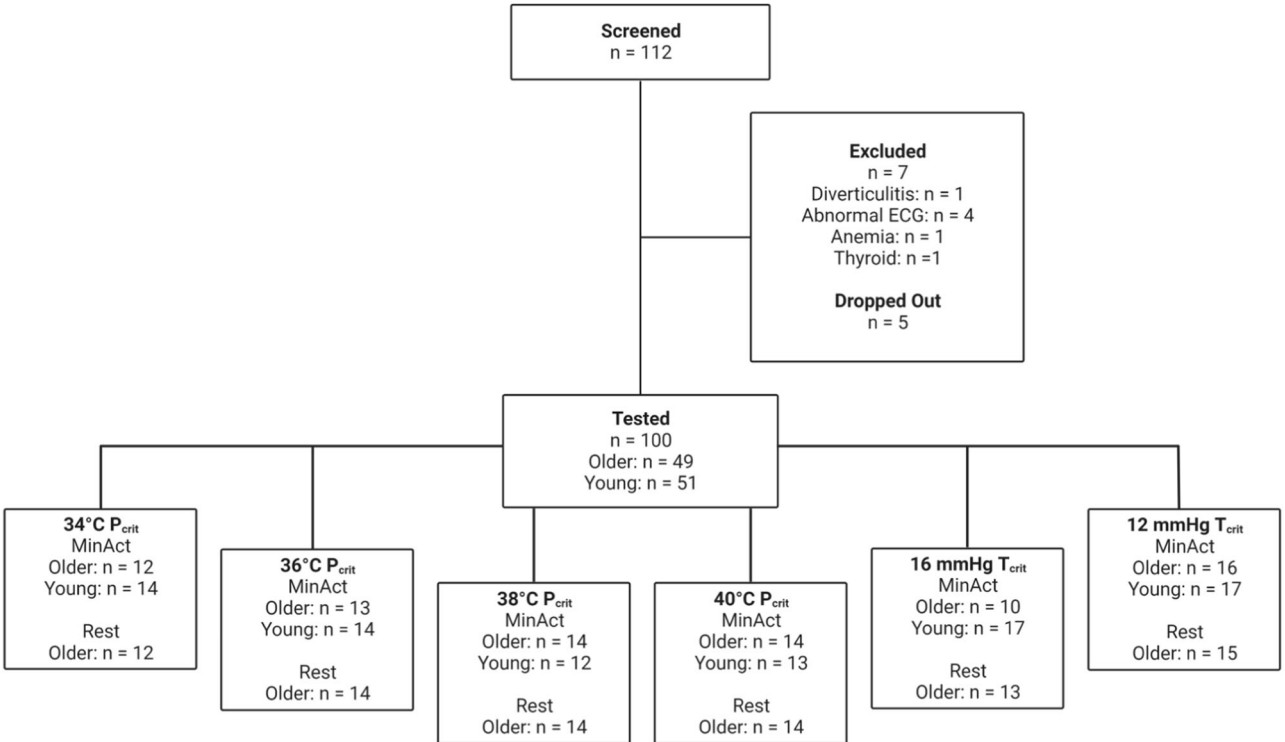

**Fig. 1 Enrollment flow chart.** Flow chart documenting the number of participants screened for the study, the number of participants excluded and reasons for exclusion, and the number of participants tested and randomized into each experimental condition. Created with BioRender.com.

($p < 0.001$), but there were no differences in height, body mass index (BMI), body surface area ($A_D$), or body surface area-to-mass ratio ($A_D \cdot kg^{-1}$) between age groups (all $p \geq 0.30$).

**Metabolic and sweat rates.** During experimental trials, oxygen consumption ($\dot{V}O_2$) was measured and net metabolic heat production ($M_{net}$) and metabolic equivalents (METs) were calculated. During MinAct, there were no differences by age group or environmental condition for $M_{net}$ ($p \geq 0.06$), $\dot{V}O_2$ ($p \geq 0.07$), or METs ($p \geq 0.07$; Table 2A). We also calculated percent body mass loss and sweat rates from the difference in body mass from pre- to post-experiment. There was a main effect of age for percent body mass loss ($p < 0.0001$) and sweat rate ($p = 0.0003$), but there were no effects of environmental condition ($p \geq 0.24$) nor interaction effects ($p \geq 0.43$).

As expected, in the older adults, $M_{net}$, $\dot{V}O_2$, METs, sweat rate, and percent body mass loss were lower during Rest (Table 2B) compared with MinAct (main effect, all $p < 0.01$), but there were no effects of environmental condition ($p \geq 0.39$) nor interaction effects ($p \geq 0.27$).

**Compensability analyses.** We used a progressive heat stress protocol to determine the point at which heat stress transitions from compensable to uncompensable (Fig. 2; see "Methods" section for details). Compensability curves (Fig. 3), modeled after traditional survival curves, were constructed to demonstrate the percentage of adults for whom heat stress was compensable (i.e., heat balance can be maintained) at a given water vapor pressure [$P_a$; critical $P_a$ ($P_{crit}$) trials, left of the dashed line] or $T_{db}$ [critical $T_{db}$ ($T_{crit}$) trials, right of the dashed line]. Compensability curves comparing young vs. older adults during MinAct are depicted in Fig. 3a. In all environmental conditions, the curves were significantly different between age groups (all $p < 0.0001$).

Figure 3b presents compensability curves demonstrating the percentage of older adults for whom heat stress became uncompensable at a given $P_a$ ($P_{crit}$ trials; left of the dashed line) or $T_{db}$ ($T_{crit}$ trials; right of the dashed line) during MinAct compared with Rest trials. The curves were significantly different between MinAct and Rest in all environmental conditions ($p \leq 0.004$) except 34 °C $P_{crit}$ ($p = 0.88$).

**Critical environmental limits.** Mean critical environmental loci were plotted on a standard psychrometric chart along with the lower bounds of the 95% CI. Heat stress is compensable in environmental conditions that are below and to the left of each line. Figure 4a depicts critical environmental loci for young and older adults during MinAct. Psychrometric limits were shifted downward and leftward across all environmental conditions in older compared with young adults ($p < 0.0001$). Summary data for $T_{db}$, $P_a$, and relative humidity (rh) at the critical environmental limits for young and older adults during MinAct trials are presented in Table 3A. The lower %rh in young adults during $T_{crit}$ trials is a function of higher $T_{db}$ at the same $P_a$.

Critical environmental loci, with lower bounds of the 95% CI, for older adults during MinAct and Rest are presented in Fig. 4B. Psychrometric limits during Rest were shifted upward and

---

**Table 1 Subject characteristics.**

| Characteristic | Older | Young |
|---|---|---|
| $n$ | 49 (21 M, 28 F) | 51 (22 M, 29 F) |
| Age (year) | 71 ± 6* | 23 ± 4 |
| Height (m) | 1.68 ± 0.1 | 1.72 ± 0.1 |
| BMI (kg•m$^{-2}$) | 26 ± 5 | 25 ± 4 |
| $A_D$ (m$^2$) | 1.82 ± 0.23 | 1.84 ± 0.20 |
| $A_D$•kg$^{-1}$ (m$^2$•kg$^{-1}$) | 0.025 ± 0.003 | 0.026 ± 0.002 |
| $\dot{V}O_{2max}$ (ml•kg$^{-1}$•min$^{-1}$) | 28 ± 9* | 49 ± 12 |

*$p < 0.05$ compared to Young.
$A_D$ DuBois body surface area, $A_D$•kg$^{-1}$ body surface area-to-mass ratio, *BMI* body mass index, $\dot{V}O_{2max}$ maximal oxygen consumption.

---

**Table 2 Metabolic heat production ($M_{net}$), oxygen consumption ($\dot{V}O_2$), metabolic equivalents (METs), sweat rates (SR), and percent body mass loss (BML) for each experimental condition (mean ± SD).**

| | 34 °C | 36 °C | 38 °C | 40 °C | 16 mmHg | 12 mmHg |
|---|---|---|---|---|---|---|
| **MinAct (older vs. young)** | | | | | | |
| $M_{net}$ (W • m$^{-2}$) | | | | | | |
| Older | 75.5 ± 9.4 | 80.7 ± 14.5 | 82.4 ± 17.6 | 84.4 ± 15.1 | 78.7 ± 15.8 | 77.8 ± 10.0 |
| Young | 85.7 ± 9.4 | 78.0 ± 13.5 | 81.9 ± 13.5 | 80.1 ± 8.7 | 85.5 ± 10.8 | 88.9 ± 10.6 |
| $\dot{V}O_2$ (L•min$^{-1}$) | | | | | | |
| Older | 0.43 ± 0.08 | 0.41 ± 0.10 | 0.48 ± 0.16 | 0.47 ± 0.13 | 0.45 ± 0.13 | 0.43 ± 0.09 |
| Young | 0.50 ± 0.07 | 0.42 ± 0.10 | 0.47 ± 0.10 | 0.46 ± 0.09 | 0.48 ± 0.09 | 0.51 ± 0.11 |
| METs (a.u.) | | | | | | |
| Older | 1.64 ± 0.17 | 1.74 ± 0.20 | 1.73 ± 0.27 | 1.86 ± 0.26 | 1.67 ± 0.27 | 1.71 ± 0.24 |
| Young | 1.72 ± 0.19 | 1.72 ± 0.27 | 1.70 ± 0.19 | 1.76 ± 0.25 | 1.86 ± 0.23 | 1.93 ± 0.23 |
| SR (g • m$^{-2}$ • h$^{-1}$) | | | | | | |
| Older* | 77.2 ± 45.1 | 84.3 ± 49.5 | 89.3 ± 57.3 | 90.4 ± 49.6 | 82.6 ± 70.0 | 92.0 ± 61.9 |
| Young | 121.8 ± 60.0 | 136.4 ± 88.6 | 202.4 ± 113.7 | 167.1 ± 69.9 | 141.7 ± 68.1 | 185.8 ± 110.0 |
| BML (%) | | | | | | |
| Older* | 0.38 ± 0.22 | 0.37 ± 0.25 | 0.39 ± 0.26 | 0.41 ± 0.25 | 0.33 ± 0.31 | 0.38 ± 0.27 |
| Young | 0.54 ± 0.30 | 0.60 ± 0.43 | 0.94 ± 0.65 | 0.89 ± 0.41 | 0.62 ± 0.31 | 0.77 ± 0.45 |
| **Rest (Older Only)** | | | | | | |
| | 34 °C | 36 °C | 38 °C | 40 °C | 16 mmHg | 12 mmHg |
| $M_{net}$ (W•m$^{-2}$)† | 48.4 ± 9.5 | 46.8 ± 7.4 | 50.2 ± 9.6 | 49.8 ± 8.0 | 52.5 ± 9.3 | 51.5 ± 9.1 |
| $\dot{V}O2$ (L•min$^{-1}$)† | 0.27 ± 0.06 | 0.25 ± 0.05 | 0.30 ± 0.08 | 0.28 ± 0.08 | 0.30 ± 0.08 | 0.29 ± 0.07 |
| METs (a.u.)† | 1.13 ± 0.26 | 1.05 ± 0.19 | 1.05 ± 0.18 | 1.14 ± 0.15 | 1.16 ± 0.15 | 1.12 ± 0.16 |
| SR (g•m$^{-2}$•h$^{-1}$)† | 44.6 ± 21.0 | 57.7 ± 40.0 | 68.2 ± 58.0 | 65.5 ± 56.3 | 51.6 ± 24.5 | 52.2 ± 31.3 |
| BML (%)† | 0.19 ± 0.10 | 0.27 ± 0.17 | 0.35 ± 0.32 | 0.31 ± 0.27 | 0.22 ± 0.09 | 0.26 ± 0.17 |

* denotes main effect $p < 0.05$ for older compared to young during MinAct; † denotes main effect $p < 0.05$ for rest compared to MinAct in older adults.

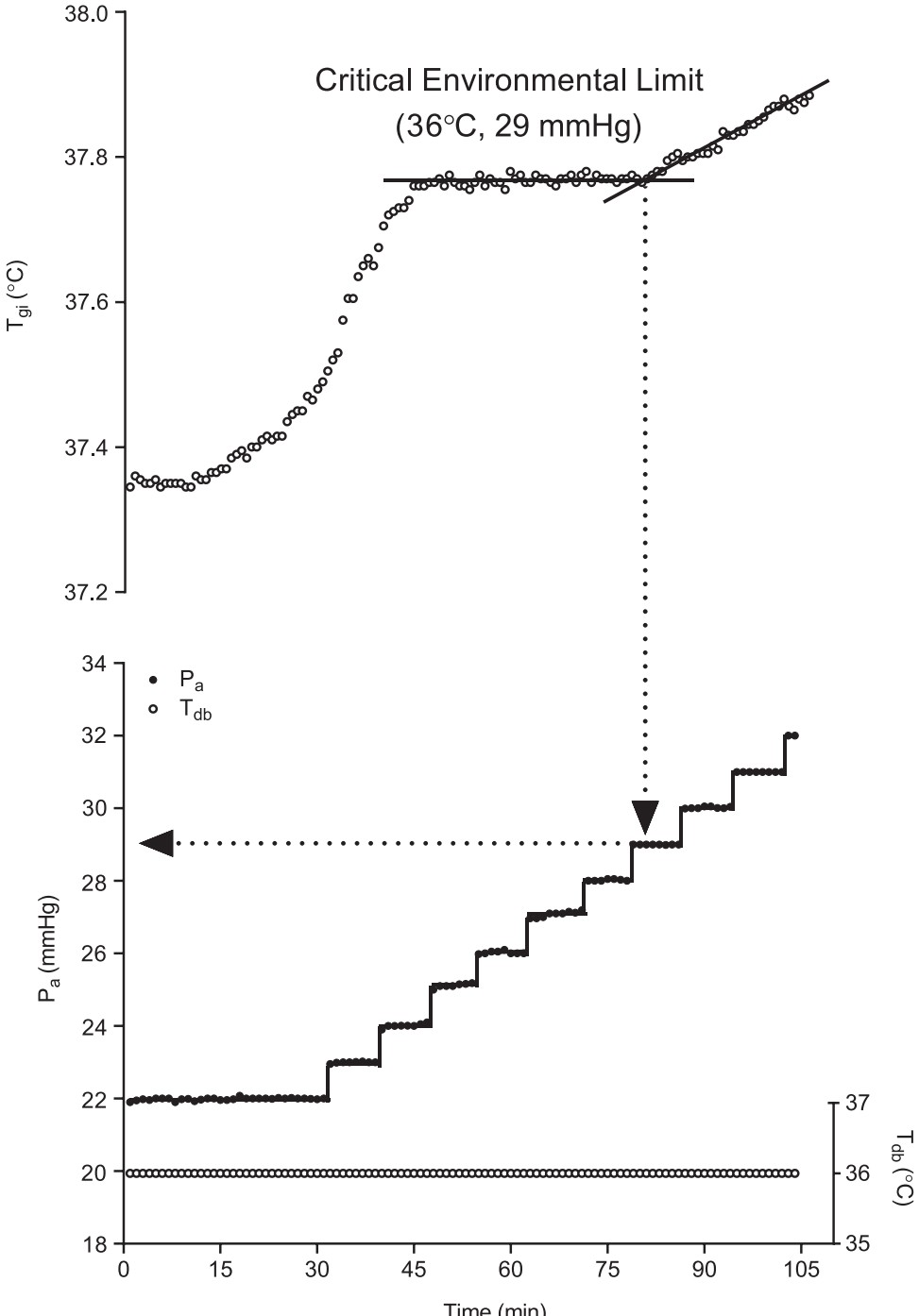

**Fig. 2 Representative tracing of the experimental protocol.** Time course of core temperature (gastrointestinal temperature, $T_{gi}$), dry-bulb temperature ($T_{db}$), and ambient water vapor pressure ($P_a$) for a Rest trial with increasing $P_a$. Lines are drawn through data points in the bottom panel to demonstrate the stepwise progression of $P_a$. The $T_{gi}$ inflection point represents the combination of environmental conditions above which heat stress becomes uncompensable and a stable core temperature can no longer be maintained. In this case, the $T_{gi}$ inflection point (i.e., critical water vapor pressure, $P_{crit}$) occurs at $P_a = 29$ mmHg.

rightward relative to MinAct (p < 0.0001). Table 3B shows summary data for $T_{db}$, $P_a$, and rh at the critical environmental limits for older adults during Rest trials.

**Core temperature rate of change.** Lastly, we calculated the rate of change in $T_{gi}$ below and above critical environmental limits. There was no effect of environment ($p \geq 0.44$) or age ($p \geq 0.14$) on the rate of change in $T_{gi}$ below (Young MinAct, 0.08 ± 0.13 °C·hr$^{-1}$; Older MinAct, 0.10 ± 0.12 °C·hr$^{-1}$; Older Rest, 0.07 ± 0.14

°C·hr$^{-1}$) or above (Young MinAct, 0.70 ± 0.54 °C·hr$^{-1}$; Older MinAct, 0.62 ± 0.48 °C·hr$^{-1}$; Older Rest, 0.69 ± 0.50 °C·hr$^{-1}$) the critical environmental limit. Slopes above the inflection point were significantly greater than below (p < 0.0001).

## Discussion
The primary goal of the ongoing PSU HEAT Project is to determine critical environmental limits for indoor settings, above which thermal equilibrium with the environment is not possible

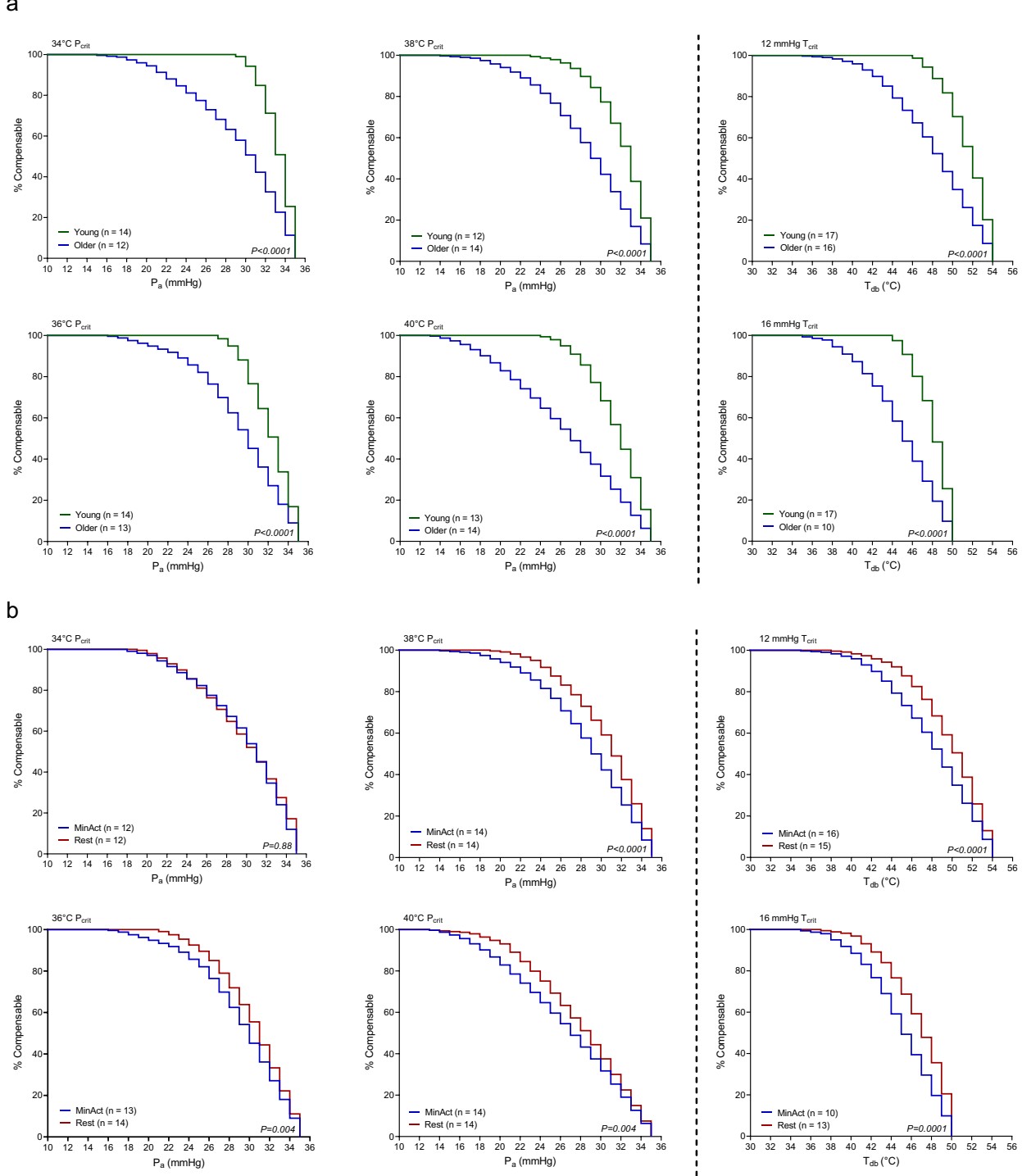

**Fig. 3 Compensability curves.** Modeled after traditional survival curves, demonstrating the percentage of adults for whom heat stress was compensable at a given water vapor pressure ($P_a$, $P_{crit}$ trials; left of the dashed line) or dry-bulb temperature ($T_{db}$, $T_{crit}$ trials; right of the dashed line). The Gehan-Breslow-Wilcoxon test was used to assess differences in survival curves. Panel A: Compensability curves for young vs. older adults during minimal physical activity (MinAct). The curves were significantly different between age groups in every environmental condition. Panel B: Compensability curves for older adults during MinAct vs. seated rest (Rest). The curves were significantly different between MinAct and Rest in every environmental condition except 34 °C $P_{crit}$.

for adults over the age of 65 yrs -- the most vulnerable age group with respect to heat-related morbidity and mortality. The present investigation conducted a compensability analysis (analogous with survival curve determination) showing excess heat vulnerability of apparently healthy older men and women compared to younger adults. Further, we established and compared critical

environmental limits for older adults during minimal physical activity associated with ADL and seated rest, and compared them to updated data from those previously-published[15,22] from a cohort of young adults. As expected, critical environmental limits for older adults during MinAct were shifted to lower combinations of ambient $T_{db}$ and humidity relative to young adults,

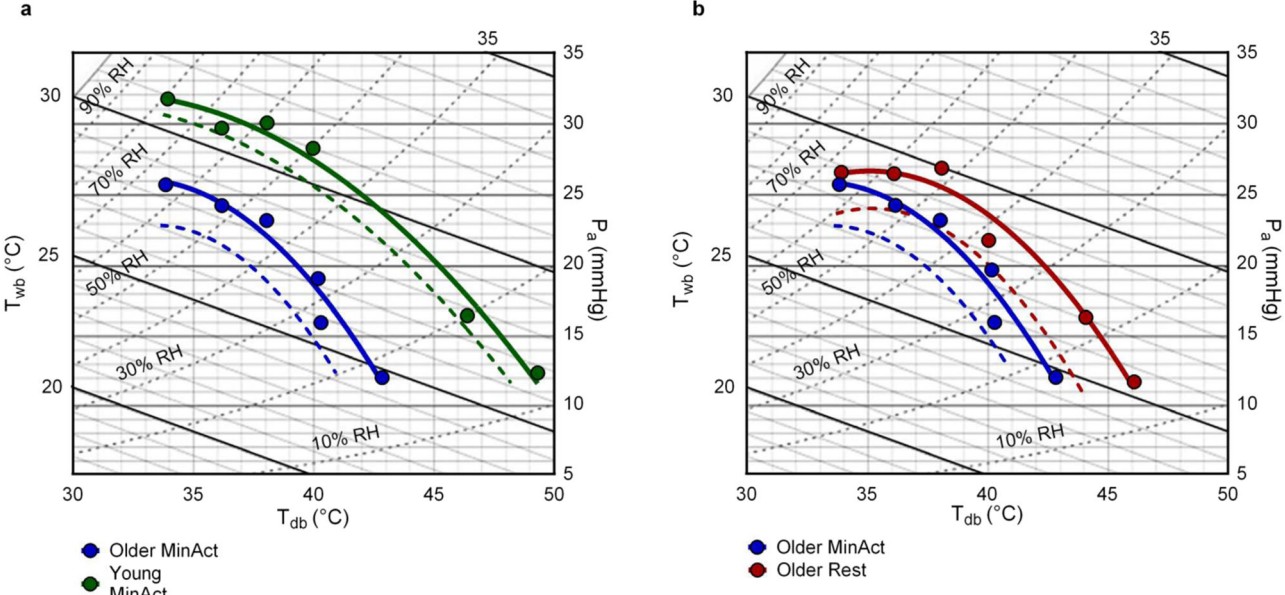

**Fig. 4 Critical environmental limits.** Standard psychrometric charts depicting empirically-derived mean critical environmental limits (symbols and solid lines). The smaller, dashed lines denote the lower bounds of the 95% confidence interval for each condition. **a** Critical environmental limits for young (green circles and lines) and older (blue circles and lines) adults during minimal physical activity (MinAct). Critical environmental limits were significantly lower (i.e., shifted downward and to the left; $P < 0.0001$) for older compared to young adults across all environmental conditions tested. **b** Critical environmental limits for older adults during minimal physical activity (MinAct; blue circles and lines) and Rest (red circles and lines). Critical environmental limits were significantly lower (i.e., shifted downward and to the left; $P < 0.0001$) during MinAct compared to Rest across all environmental conditions tested. Mixed-effects models were used to evaluate the effect of age (**a**) and metabolic rate (**b**) on psychrometric limits across all environmental conditions.

**Table 3 Environmental conditions at the core temperature inflection point for each experimental condition (mean ± 95% CI).**

|  | $T_{db}$ (°C) | $P_a$ (mmHg) | rh (%) |
|---|---|---|---|
| **MinAct (Older vs. Young)** |  |  |  |
| 34 °C |  |  |  |
| Older | 33.9 [33.7, 34.0] | 25.6 [22.6, 28.6] | 62.1 [52.8, 71.4] |
| Young | 33.9 [33.7, 34.2] | 31.7 [30.9, 32.5]* | 79.7 [77.6, 81.7]* |
| 36 °C |  |  |  |
| Older | 36.2 [36.0, 36.4] | 24.1 [21.7, 26.5] | 53.5 [48.2, 58.8] |
| Young | 36.0 [35.9, 36.1] | 29.6 [28.6, 30.7]* | 66.5 [64.1, 69.0]* |
| 38 °C |  |  |  |
| Older | 38.0 [37.8, 38.3] | 23.0 [20.6, 25.5] | 46.1 [41.7, 50.6] |
| Young | 38.1 [37.8, 38.4] | 29.7 [27.8, 31.6]* | 59.5 [55.5, 63.6]* |
| 40 °C |  |  |  |
| Older | 40.2 [40.0, 40.3] | 19.0 [16.3, 21.7] | 34.0 [29.2, 38.8] |
| Young | 40.0 [39.9, 40.1] | 28.2 [26.6, 29.7]* | 51.0 [48.2, 53.8]* |
| 16 mmHg |  |  |  |
| Older | 40.3 [38.4, 42.2] | 15.9 [15.6, 16.2] | 28.6 [25.8, 31.5] |
| Young | 46.4 [45.5, 47.3]* | 16.3 [16.0, 16.5] | 20.7 [19.9, 21.5]* |
| 12 mmHg |  |  |  |
| Older | 42.9 [40.9, 44.8] | 11.9 [11.6, 12.2] | 18.8 [17.0, 20.5] |
| Young | 49.3 [48.2, 50.4]* | 12.2 [12.0, 12.4] | 13.8 [12.9, 14.7]* |
| **B. Rest (Older Only)** |  |  |  |
|  | $T_{db}$ (°C) | $P_a$ (mmHg) | rh (%) |
| 34 °C | 33.9 [33.7, 34.1] | 26.4 [22.7, 30.2] | 66.5 [55.9, 77.1] |
| 36 °C | 36.2 [35.9, 36.4] | 26.4 [24.4, 28.4] | 59.1 [54.6, 63.5] |
| 38 °C | 38.1 [37.9, 38.3] | 26.7 [24.2, 29.3]† | 53.4 [48.4, 58.5]† |
| 40 °C | 40.1 [40.0, 40.1] | 21.6 [19.3, 24.0] | 38.8 [34.5, 43.1] |
| 16 mmHg | 44.1 [41.8, 46.4]† | 16.1 [15.6, 16.5] | 24.0 [21.2, 26.8] |
| 12 mmHg | 46.1 [44.1, 48.1]† | 11.5 [11.2, 11.9] | 15.6 [13.7, 17.5] |

*$P < 0.05$ compared to older; †$P < 0.05$ compared to MinAct.
$T_{db}$ dry-bulb temperature, $P_a$ water vapor pressure, rh relative humidity, MinAct minimal activity.

demonstrating a constrained range of heat-compensable environments for older adults. On the other hand, environmental loci were shifted to higher temprature/humidity combinations in older adults at rest relative to MinAct.

Our laboratory and others have identified critical environmental limits for prolonged heat exposure across a wide range of environments, at different metabolic rates, and in various clothing ensembles[9–12,14,16]. However, most of those studies have been conducted in healthy young adults and at higher metabolic rates, i.e., those associated with industry and sport. Only one study previously examined critical environmental limits for adults >60 yr of age[14]. That study compared critical environmental limits in unacclimated older women to those of heat-acclimated and unacclimated young women during treadmill walking at an intensity equal to 30% $VO_{2max}$. That intensity was historically used in these studies because it is the intensity associated with an 8-h work day in many industrial settings[24] and it reflects the intensity of many self-paced recreational activities. However, in the context of protecting the most vulnerable adults during extreme heat events, those metabolic rates are not typically experienced by most older adults who succumb during extreme heat events[21,25,26]. Thus, the present investigation tested older adults during Rest and MinAct; two low metabolic rates that better reflect those likely to be experienced by older adults during heat waves.

Heat stress vulnerability, as defined by a relative inability to keep core temperature from rising progressively, can be depicted by constructing "survival-style" heat stress compensability curves, showing the percentage of adults within each group for whom heat stress became uncompensable across the range of environments. Unsurprisingly, the curves were different between young and older adults during MinAct, suggesting that heat stress is uncompensable for a larger pecentage of older adults at each combination of $T_{db}$ and $P_a$ relative to young adults. Similarly, heat stress was typically uncompensable for a larger pecentage of older adults at lower combinations of $T_{db}$ and $P_a$ during MinAct relative to Rest.

When the critical limits for heat balance are plotted on a psychrometric chart, the curvilinear relation between $P_a$ and $T_{db}$ is explained by the transition from fully wetted skin at the upper-left portion of the curve to free evaporation at the lower-right portion of the curve. The slope is constrained in the upper left portion of the curve by a limited capacity for evaporative cooling, secondary to a relatively small $P_a$ gradient between the skin and the environment. Hotter, drier ambient environments (i.e., the lower-right aspect of the curve) allow for free evaporation of sweat; thus, the limit to heat balance is defined by maximal sweating capacity, resulting in a more vertical slope. As demonstrated by Fig. 4A, the slope of the line in hot, dry ambient conditions was steeper in older compared to young adults. The relatively wider and more linear nature of that portion of the curve in young compared to older adults can be explained by higher sweating rates in young adults[27–30]. Likewise, that portion of the curve was wider and more linear during Rest compared to MinAct in older adults (Fig. 4B), due to lower metabolic rates and, therefore, a lower requirement for sweat evaporation to maintain heat balance.

Similar to our previously-published data in young adults[22], we calculated the rate of change in $T_{gi}$ immediately below and above the critical environmental limits. In those environments that immediately precede the $T_{gi}$ inflection point, a slightly positive rate of change in $T_{gi}$ (~0.07–0.10 °C·min⁻¹) constitutes minimal rates of heat storage as heat stress approaches uncompensability. Above the critical environmental limits, the rate of change was significantly greater. Those findings were consistent, regardless of environment (i.e., warm-humid or hot-dry), metabolic rate

(MinAct vs. Rest), or age, and consistent with our previous study[22]. Along with our empirically-derived critical environmental limits, these data can be used make projections from and to any hypothetical $T_{gi}$ in conditions that are either below or above those limits.

**Potential limitations**. No attempt was made to control for heat acclimatization in this study. However, subjects were tested throughout the calendar year with enough participants tested during each season to eliminate any potential influence of natural acclimatization status. The present study tested a cohort of participants residing in central Pennsylvania; it is conceivable that critical environmental limits may be shifted to higher combinations of $T_{db}$ and humidity for populations living in regions with more thermally stressful environments. Additionally, subjects were tested in an environmental chamber with no source of radiative heat transfer and minimal air movement; thus, the environmental limits presented in this study may not extend to outdoor settings.

Older adults in this study were recruited without regard to medical history, blood pressure, blood biochemistry, medications, etc., as long as there were no safety contraindications to their participation. This recruitment approach was used to increase generalizability of the results to the population for these age groups. It remains to be examined whether overt comorbidities (e.g., heart disease, diabetes) or medications (e.g., diuretics, statins, etc.) influence the upper limits for the maintenance of heat balance.

This study was not powered to assess potential sex differences in critical environmental limits. Additionally, we did not control for menstrual status or contraceptive use in young women. A long-term goal of the PSU HEAT project is to test enough participants to determine whether sex differences exist, with a large enough sample of young women that we can establish safe environmental limits that are independent of menstral cycle phase or contraceptive use.

**Conclusions and perspectives**. The present investigation establishes heat-related vulnerability in terms of critical environmental limits for older men and women during seated rest, as well as at an energy expenditure associated with activities of daily living, in an indoor setting. As expected, moving from rest to minimal activity shifted the critical psychrometric loci downward and leftward, such that a narrower range of environments was physiologically compensable. Similarly, during minimal activity, critical environmental limits were reduced in older adults and a smaller percentage of older adults was able to maintain a stable $T_{gi}$ in progressively stressful thermal indoor environments relative to their young counterparts. Heat balance cannot be maintained in those indoor environments that are above the critical environmental limit for a given metabolic rate, thus resulting in a continuous rise in core temperature and potentially increasing risk of heat-related illnesses. As such, the data from this study can be used to develop evidence-based guidelines and safety interventions to protect the particularly vulnerable population of adults over the age of 65 yrs during extreme heat events.

## Methods
All experimental procedures are registered on ClinicalTrials.gov (NCT0428439), received approval from the Institutional Review Board at The Pennsylvania State University, and conformed to the guidelines set forth by the Declaration of Helsinki. After all aspects of the experimental procedures were explained, oral and written informed consent were obtained. One-hundred and twelve subjects (54 young, 58 older) were recruited and screened

(see Fig. 1) from the Centre County region in Pennsylvania; seven were excluded from participation and five dropped out before completing an experimental trial. Fifty-one young ($23 \pm 4$ yrs) and 49 older ($71 \pm 6$ yrs) adults were randomly assigned to complete experimental trials in two environmental conditions during MinAct (young and older subjects) and seated rest (Rest; older adults only). Participants, but not investigators, were blinded to the environmental condition in which they were being tested. MinAct data for young subjects included herein have been updated from previously-published data[15,22] and are included for comparison with novel data from older subjects. Because each subject completed two to four experimental trials, a total of 248 trials were completed among the 100 subjects who were tested.

To increase generalizability of the results to the general population, subjects were recruited without regard to body size, blood pressure, blood biochemistry, etc. Subjects were excluded if they had any contraindications to heat exposure or low-intensity physical activity, mobility limitations, abnormal resting or exercise electrocardiogram (all older subjects were cleared for participation by a cardiologist), a history of Crohn's disease, diverticulitis, or similar gastrointestinal disorder, used tobacco or illegal/recreational drugs, were pregnant or planning to become pregnant, or if they were taking any medications that may render them unsafe to perform physical activity in the heat. No attempt was made to control for menstrual status or contraceptive use in young women. $\dot{V}O_{2max}$ was determined with the use of open-circuit spirometry (Parvo Medics TrueOne® 2400, Parvo, UT, USA) during a maximal graded exercise test performed on a motor-driven treadmill. During the experiments, clothing was standardized to thin, short-sleeved cotton tee-shirts, a sports bra (women), shorts, socks, and walking/running shoes.

**Testing procedures**. Experimental trials were conducted on separate days with at least 72 h between visits. Participants were asked to abstain from caffeine consumption for at least 12 h and alcohol consumption or vigorous exercise for 24 h prior to arrival for experimental trials. Upon arrival at the laboratory, participants provided a urine sample to ensure euhydration, defined as urine specific gravity ≤ 1.020 (USG; PAL-S, Atago, Bellevue, WA, USA)[31]. Young and older subjects cycled on a recumbent cycle ergometer (Lode Excalibur, Groningen, The Netherlands) in an environmental chamber against zero resistance at a cadence of 40-50 rpm, a rate that reflected the metabolic demand of minimal physical activity associated with ADL (MinAct trials)[32,33]. Older adults were also tested at rest in the same environmental conditions as those in which they performed MinAct trials.

During $P_{crit}$ trials, $T_{db}$ was held constant at either 34, 36, 38, or 40 °C. Conversely, during $T_{crit}$ trials, $P_a$ was held constant at either 12 or 16 mmHg. These experimental conditions represent a range of environments from warm-humid to hot-dry ($T_{db}$ range of 34–~50 °C and relative humidity range of ~10−80%). After a 30-min equilibration period, either the $T_{db}$ ($T_{crit}$ tests) or the $P_a$ ($P_{crit}$ tests) in the environmental chamber was increased in a stepwise fashion (1°C or 1 mmHg every 5 min). During each experiment, subjects continuously free-pedaled (MinAct) or remained seated (Rest) until a clear, sustained rise in $T_{gi}$ above an equilibrium temperature was observed. Total experiment time was typically 90–120 min. There was no forced air movement in the chambers; air velocity was <2 m.sec$^{-1}$.

**Measurements**. Gastrointestinal temperature telemetry capsules (VitalSense, Philips Respironics, Bend, OR, USA; BodyCap, Hérouville-Saint-Clair, France) were ingested 1-2 h before reporting to the laboratory[15,34]. $T_{gi}$ and HR data were continuously measured and recorded. We have previously demonstrated excellent concurrence between rectal and gastrointestinal temperatures for the determination of the $T_c$ inflection point (ICC = 0.93)[13]. Oxygen consumption ($\dot{V}O_2$; L/min) and respiratory exchange ratio (RER; unitless) were measured at 5 and 60 min into the experimental trial.

Partitional calorimetry was used to calculate $M_{net}$ as described previously[15,35]. To compare work rates in terms of METs to previous data[33] detailing the metabolic costs of various household tasks, METs were calculated using $\dot{V}O_2$ in mL·kg$^{-1}$·min$^{-1}$, assuming a resting $\dot{V}O_2$ of 3.5 mL·kg$^{-1}$·min$^{-1}$ [15,36].

Sweat rate (in g·m$^{-2}$·h$^{-1}$) was calculated after each experiment from the loss of nude body mass (i.e., the difference in body mass from pre- to post-experiment) on a scale accurate to ±10 g. For clarity, both percent body mass loss and sweat rate are reported. Fluid intake was not provided between the initial and final measurements of nude body mass.

**Determination of $T_{crit}$ and $P_{crit}$**. The critical $T_{db}$ or $P_a$ were determined for each trial as previously described[11,14,15]. Briefly, after an initial rise, $T_{gi}$ typically began to plateau after 30–60 min and remained at an elevated, relative steady state, as $T_{db}$ or $P_a$ was systematically increased in a stepwise fashion. The critical $T_{db}$ or $P_a$ was characterized by initiation of a subsequent, sustained upward inflection of $T_{gi}$ from steady state, which was selected graphically from the raw data by drawing a first line between the data points after the $T_{gi}$ plateau and a second line from the point of departure from the $T_{gi}$ equilibrium phase slope (Fig. 2). The average $T_{db}$ or $P_a$ for the 2 min immediately preceding the inflection point was defined as the critical $T_{db}$ or $P_a$, respectively. Inflection points determined by visual inspection and using segmental linear regression analysis were in excellent agreement (ICC = 0.99)[37]. We have additionally demonstrated excellent intra-rater reliability for the determination of the $T_c$ inflection point (ICC = 0.93)[13].

During 34 °C $P_{crit}$ trials, no $T_{gi}$ inflection was observed after more than 120 min for three subjects during Rest trials and for two subjects during MinAct trials. In those cases, heat balance equations were used[11,14,15] to construct individual isothermal lines, and the slope of the line was used to estimate that individual's inflection point. In these few cases, the estimated inflection point was <5 mmHg higher than that during the final stage completed and fit within the range of measured age-group values with no outliers.

**Data analysis and statistics**. An a priori power analysis using an effect size of 1.2, based on previously published $P_{crit}$ data[14,16], suggested that seven subjects per group would yield sufficient statistical power (power ≥ 0.8, α = 0.05) to detect significant differences in $T_{crit}$ and $P_{crit}$.

Subject characteristics (height, BMI, $A_D$, and $A_D$·kg$^{-1}$) were compared between age groups using independent-samples t tests (SPSS, version 9.4, IBM, Armonk, NY). Mixed-effects models were conducted to compare $M_{net}$, $\dot{V}O_2$, METs, sweat rate, and total body mass loss between age groups, metabolic rates, and among environmental conditions.

Compensability curves (modeled after traditional survival curves) were constructed (GraphPad Prism, v. 9.2, GraphPad Software, San Diego, CA) to demonstrate the percentage of adults for whom heat stress was compensable at a given $P_a$ ($P_{crit}$ trials) or $T_{db}$ ($T_{crit}$ trials) for young vs. older adults (MinAct trials) and for older adults during MinAct vs. Rest in each environmental condition. The Gehan-Breslow-Wilcoxon test was used to assess differences in survival curves.

Psychrometric loci were plotted on a standardized psychrometric chart from the mean data for combinations of $T_{db}$ and $P_a$.

Psychrometric curves for the mean and 95% CI data were constructed using a second-order least squares regression polynomial curve fitting procedure. Separate mixed-effects models were used to evaluate 1) the effect of age (young vs. older) on psychrometric limits across all environmental conditions during MinAct, 2) the effect of metabolic rate (MinAct vs. Rest) on psychrometric limits across all environmental conditions within older adults, and 3) differences in the rate of change in $T_{gi}$ above and below critical environmental limits between age groups across environments.

Significance was accepted for all analyses at p = 0.05. Data are presented as mean ± SD except in Fig. 4 and Table 3 which are presented as mean data and 95% confidence intervals (CI).

**Data sharing**. Source data supporting the conclusions in this paper are publicly available and can be found at https://zenodo.org/records/10197956.

**Reporting summary**. Further information on research design is available in the Nature Portfolio Reporting Summary linked to this article.

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

### Acknowledgements

This work was supported by NIH Grant R01 AG067471 (WLK), NIA Grant T32 AG049676 (RMC and DJV), and an American College of Sports Medicine Foundation doctoral student research grant (RMC). The authors are thankful for the subject's participation and for the assistance of Susan Slimak, RN.

### Author contributions

WLK and STW designed the study; STW, RMC, KGF, and DJV conducted research; STW and WLK interpreted results of experiments; STW analyzed data; STW, RMC, and KGF directly accessed and verified the data reported in the manuscript; STW drafted manuscript; STW, RMC, KGF, DJV, and WLK revised and edited manuscript; STW,

RMC, KGF, DJV, and WLK approved final version of manuscript, and accept responsibility to submit for publication.

## Competing interests

The authors declare no competing interests.
