## [Peer Review File · Communications Earth & Environment]

Decision letter and referee reports: first round

10th Oct 23

Dear Dr Wolf,

Your manuscript titled "Heat stress vulnerability and critical environmental limits for older adults (PSU HEAT Project)" has now been seen by 2 reviewers, and we include their comments at the end of this message. They find your work of interest, but some important points are raised. We are interested in the possibility of publishing your study in Communications Earth & Environment, but would like to consider your responses to these concerns and assess a revised manuscript before we make a final decision on publication.

We therefore invite you to revise and resubmit your manuscript, along with a point-by-point response that takes into account the points raised. Please highlight all changes in the manuscript text file.

Please use the following link to submit your revised manuscript, point-by-point response to the referees' comments (which should be in a separate document to any cover letter), a tracked-changes version of the manuscript (as a PDF file) and the completed checklist:

[Link redacted]

We hope to receive your revised paper within six weeks; please let us know if you aren't able to submit it within this time so that we can discuss how best to proceed. If we don't hear from you, and the revision process takes significantly longer, we may close your file. In this event, we will still be happy to reconsider your paper at a later date, as long as nothing similar has been accepted for publication at Communications Earth & Environment or published elsewhere in the meantime.

Please do not hesitate to contact us if you have any questions or would like to discuss these revisions further. We look forward to seeing the revised manuscript and thank you for the opportunity to review your work.

Best regards,

Clare Davis, PhD
Senior Editor
Communications Earth & Environment

www.nature.com/commsenv/
@CommsEarth

EDITORIAL POLICIES AND FORMATTING

Editorial Policy: [Policy requirements](https://www.nature.com/documents/nr-editorial-policy-checklist.pdf) (Download the link to your computer as a PDF.)

Furthermore, please align your manuscript with our format requirements, which are summarized on the following checklist:

[Communications Earth & Environment formatting checklist](https://www.nature.com/documents/commsj-phys-style-formatting-checklist-article.pdf)

and also in our style and formatting guide [Communications Earth & Environment formatting guide](https://www.nature.com/documents/commsj-phys-style-formatting-guide-accept.pdf).

*** DATA: Communications Earth & Environment endorses the principles of the Enabling FAIR data project (<http://www.copdess.org/enabling-fair-data-project/>). We ask authors to make the data that support their conclusions available in permanent, publically accessible data repositories. (Please contact the editor if you are unable to make your data available).

All Communications Earth & Environment manuscripts must include a section titled "Data Availability" at the end of the Methods section or main text (if no Methods). More information on this policy, is available at <http://www.nature.com/authors/policies/data/data-availability-statements-data-citations.pdf>.

If a community resource is unavailable, data can be submitted to generalist repositories such as [figshare](https://figshare.com/) or [Dryad Digital Repository](http://datadryad.org/). Please provide a unique identifier for the data (for example a DOI or a permanent URL) in the data availability statement, if possible. If the repository does not provide identifiers, we encourage authors to supply the search terms that will return the data. For data that have been obtained from publically available sources, please provide a URL and the specific data product name in the data availability statement. Data with a DOI should be further cited in the methods reference section.

REVIEWER COMMENTS:

Reviewer #1 (Remarks to the Author):

GENERAL COMMENTS

Thank you for the opportunity to review your manuscript. The subject matter of the paper is compelling and aligns well with the journal's scope. The writing is lucid and the composition is commendable. Indeed, the manuscript holds value as it presents both novel and practically relevant information.

SPECIFIC COMMENTS

L 47-49: Temperature and humidity, although of great importance, are not the only environmental factors that play a critical role in human thermoregulation. This is especially true in light of recent studies by the HEAT-SHIELD consortium in the EU, which show that often 30°C under the sun is worse than 40°C in the shade.

L 80: These are likely the essential environmental thresholds for indoor conditions. For outdoor settings, solar radiation is probably the primary environmental factor influencing physiological heat strain.

L 89-91: Given that the current study, of exceptional quality, primarily addresses temperature and humidity, it's essential to ensure clarity throughout the paper. Especially for non-expert readers, it should be evident that the study focuses exclusively on indoor conditions. Consequently, please consider rephrasing to: 'heat-related morbidity and mortality under indoor conditions...'

L 96-97: "across adverse indoor environmental conditions"

L 106-108: The authors deserve commendation for securing a surprisingly large sample size, which enhances the generalizability of the findings!

L 109-112: Do the authors possess measurements of body composition, such as DXA or something similar? If so, I would recommend including data on 'body fat' and the ratio of body surface area to lean body mass.

L 129-130: I would recommend either changing the term "temperature" to "dry-bulb temperature" or vice versa. Consistency throughout the paper is crucial, particularly for non-experts who will be reading your work.

L 159-160: It's crucial to specify that these findings pertain solely to indoor conditions and do not necessarily apply to outdoor settings, especially under direct sunlight.

L 161: Furthermore, I'd recommend a slight rewording of "the most vulnerable population with respect to heat-related morbidity and mortality" to enhance clarity. While I don't necessarily disagree with the authors, it's crucial to highlight that obtaining large-scale epidemiological morbidity/mortality data for individuals with underlying conditions affecting human thermoregulatory capacity (e.g., abnormal blood pressure, diabetes, cystic fibrosis, etc.) can be exceedingly challenging, if not impossible. Young individuals with such conditions might, in fact, be more susceptible than their older, healthy counterparts.

L 185: It's essential to clarify that a "heat wave" is defined as a unique weather event characterized by abnormally high temperatures sustained for a period of at least 72 hours. Given that the current

study did not simulate a "heat wave" and its results don't capture the nuances of such extreme events, I'd recommend adjustments in phrasing throughout the manuscript to ensure clarity.

L 217: I believe the authors overlooked a pivotal limitation of the study. The determined critical environmental limits are applicable solely to indoor settings or conditions where the effects of solar radiation and wind speed are minimal. These results don't extend to typical outdoor scenarios. It's crucial for the authors to highlight this distinction for readers. I'd also advise reviewing recent studies from Denmark, Greece, and the UK that delve into the impacts of solar radiation on human physiology and cognition.

L 218: This underscores the importance of avoiding references to "heat wave." The physiological impacts of heat on humans during a heat wave are profoundly magnified, primarily because individuals aren't acclimatized to such elevated temperatures.

L 234-235: It is vital to avoid making such definitive statements throughout the manuscript. Similar studies have previously been conducted, involving substantial sample sizes and direct calorimetry measurements. For instance, consider the study with 167 healthy participants, aged 31-70, recently published in the journal *Temperature*: <https://doi.org/10.1080/23328940.2017.1381800> . I would suggest discussing the aforementioned study.

L 244-247: Referencing my earlier comments, it's crucial to clarify for the reader that these guidelines are intended solely for indoor conditions.

L 286-288: I recommend computing a thermal stress indicator, such as the Wet-Bulb Globe Temperature, and considering further analyses using this calculated metric. To aid in this, you can utilize a freely available software detailed in a recent systematic review for computing all meteorology-based thermal stress indicators: <https://doi.org/10.1080/23328940.2022.2037376>

L 306-307: As this represents a rate and not total body water loss, it should be articulated as sweat loss per unit of body surface area per minute (e.g., mg/cm²/min). If this isn't the intended representation, I'd suggest rephrasing to "body water loss" throughout the manuscript.

Reviewer #2 (Remarks to the Author):

The manuscript titled "Heat stress vulnerability and critical environmental limits for older adults" evaluated older adults' responses to a range of environmental conditions during seated rest and activity that represents typical daily living. The active responses were also compared to younger adults. These findings are the first to document age dependent vulnerability to adverse environmental conditions, and lower environmental limits in older compared to younger adults. The manuscript is clear, logical, and well written. The manuscript adopts robust methodological approaches that are leading within the domain of research, with large sample sizes and appropriate controls. The results are clear and well-presented, and the interpretation is logical and accurate. Overall, the manuscript provides novel insights into this domain of research and will act as an essential reference point for future research in this area. Due to the exceptional production of this manuscript, I have little to comment on and would recommend this article to be accepted for publication.

Minor comments

The sample size (N = 51 younger; N = 49 older adults) is commendable and great to see so many

females present within the sample size, however, I was left confused how your a priori power analysis led to the recruitment of such large sample sizes and whether this may have resulted in type 1 error.

Whilst the authors have acknowledged that no attempt was made to control for menstrual status or contraceptive use in young women, this is an experimental limitation, that should be addressed during the PSU HEAT projects long-term goal to determine sex differences in responses.

We would like to thank the reviewers for their comments, and express our appreciation for the opportunity to improve this manuscript. Below, we have included a point-by-point reply to reviewer comments with our responses in red text. We feel that the quality of the manuscript has improved markedly by addressing the reviewer suggestions, and we appreciate the time and effort of the reviewers on our behalf.

Reviewer #1 (Remarks to the Author):

GENERAL COMMENTS

Thank you for the opportunity to review your manuscript. The subject matter of the paper is compelling and aligns well with the journal's scope. The writing is lucid and the composition is commendable. Indeed, the manuscript holds value as it presents both novel and practically relevant information.

Thank you.

SPECIFIC COMMENTS

L 47-49: Temperature and humidity, although of great importance, are not the only environmental factors that play a critical role in human thermoregulation. This is especially true in light of recent studies by the HEAT-SHIELD consortium in the EU, which show that often 30°C under the sun is worse than 40°C in the shade.

We recognize and agree that radiative heat gain represents an important component of heat balance in outdoor conditions. The current study focused on temperature and humidity conditions, most relevant to indoor environments. We have noted that the critical environmental limits identified in this manuscript are specific to indoor environments here and throughout the manuscript.

L 80: These are likely the essential environmental thresholds for indoor conditions. For outdoor settings, solar radiation is probably the primary environmental factor influencing physiological heat strain.

The reviewer is correct that temperature and humidity are the essential metrics when considering indoor environments. Although the current manuscript focuses on indoor environments, critical environmental limits could more broadly include solar radiation and wind. Because this manuscript focuses on temperature and humidity only, we have added “e.g.” to the parenthetical reference to T_{ab} and humidity, thus allowing room to consider other important environmental metrics.

L 89-91: Given that the current study, of exceptional quality, primarily addresses temperature and humidity, it's essential to ensure clarity throughout the paper. Especially for non-expert readers, it should be evident that the study focuses exclusively on indoor conditions. Consequently, please consider rephrasing to: 'heat-related morbidity and mortality under indoor conditions...'

We appreciate this suggestion and have added a sentence to the preceding paragraph highlighting the importance for establishing temperature-humidity combinations above which heat stress cannot be maintained in indoor settings, and revised the sentence in question to say “while in the home or other indoor settings...”

L 96-97: "across adverse indoor environmental conditions"

We have revised the sentence accordingly.

L 106-108: The authors deserve commendation for securing a surprisingly large sample size, which enhances the generalizability of the findings!

Thank you.

L 109-112: Do the authors possess measurements of body composition, such as DXA or something similar? If so, I would recommend including data on 'body fat' and the ratio of body surface area to lean body mass.

Unfortunately, for this study we did not perform any measurements of body composition and are therefore not able to include these data.

L 129-130: I would recommend either changing the term "temperature" to "dry-bulb temperature" or vice versa. Consistency throughout the paper is crucial, particularly for non-experts who will be reading your work.

We have revised the manuscript to use dry-bulb temperature (T_{db}) throughout.

L 159-160: It's crucial to specify that these findings pertain solely to indoor conditions and do not necessarily apply to outdoor settings, especially under direct sunlight.

We have revised the sentence to read “...critical environmental limits for indoor settings...”

L 161: Furthermore, I'd recommend a slight rewording of "the most vulnerable population with respect to heat-related morbidity and mortality" to enhance clarity. While I don't necessarily disagree with the authors, it's crucial to highlight that obtaining large-scale epidemiological morbidity/mortality data for individuals with underlying conditions affecting human thermoregulatory capacity (e.g., abnormal blood pressure, diabetes, cystic fibrosis, etc.) can be exceedingly challenging, if not impossible. Young individuals with such conditions might, in fact, be more susceptible than their older, healthy counterparts.

We have revised “population” to “age group.”

L 185: It's essential to clarify that a "heat wave" is defined as a unique weather event characterized by abnormally high temperatures sustained for a period of at least 72 hours. Given that the current study did not simulate a "heat wave" and its results don't capture the nuances of

such extreme events, I'd recommend adjustments in phrasing throughout the manuscript to ensure clarity.

We appreciate this point made by the reviewer, and recognize that the current study did not simulate a heat wave. However, it is not suggested here, or at any other point in the manuscript, that the study represents simulated heat wave conditions. Rather, the sentence in question simply highlights that the metabolic rates tested in the current study are reflective of those likely to be experienced during heat waves, given that most older adults who succumb to extreme heat are resting or performing activities of daily living in and around the home.

L 217: I believe the authors overlooked a pivotal limitation of the study. The determined critical environmental limits are applicable solely to indoor settings or conditions where the effects of solar radiation and wind speed are minimal. These results don't extend to typical outdoor scenarios. It's crucial for the authors to highlight this distinction for readers. I'd also advise reviewing recent studies from Denmark, Greece, and the UK that delve into the impacts of solar radiation on human physiology and cognition.

We have added this limitation to the *Potential Limitations* section of the manuscript:

“Additionally, subjects were tested in an environmental chamber with no source of radiative heat transfer and minimal air movement; thus, the environmental limits presented in this study may not extend to outdoor settings.”

We appreciate the suggestion to review studies that examine the impacts of solar radiation on human physiology and cognition; however, those studies are only tangentially related to the current study and would provide limited additional insight while adding significantly to the length of the manuscript. We have therefore opted to forego any review of the potential impacts of solar radiation or increasing air movement velocity.

L 218: This underscores the importance of avoiding references to "heat wave." The physiological impacts of heat on humans during a heat wave are profoundly magnified, primarily because individuals aren't acclimatized to such elevated temperatures.

We agree with this point raised by the reviewer. However, the only references made to “heat waves” were with regard to epidemiological data in the introduction and, as previously mentioned, highlighting that the metabolic rates tested in the current study are reflective of those likely to be experienced by older adults during heat waves.

L 234-235: It is vital to avoid making such definitive statements throughout the manuscript. Similar studies have previously been conducted, involving substantial sample sizes and direct calorimetry measurements. For instance, consider the study with 167 healthy participants, aged 31-70, recently published in the journal

Temperature: <https://doi.org/10.1080/23328940.2017.1381800> . I would suggest discussing the aforementioned study.

We have revised this statement to more accurately reflect the novelty of the present study.

L 244-247: Referencing my earlier comments, it's crucial to clarify for the reader that these guidelines are intended solely for indoor conditions.

We have revised the concluding paragraph to clarify that these results are specific to indoor environments.

L 286-288: I recommend computing a thermal stress indicator, such as the Wet-Bulb Globe Temperature, and considering further analyses using this calculated metric. To aid in this, you can utilize a freely available software detailed in a recent systematic review for computing all meteorology-based thermal stress indicators: <https://doi.org/10.1080/23328940.2022.2037376>

We agree with the reviewer that heat stress indices such as wet-bulb globe temperature can be useful. Given that the current study was conducted in an indoor environment, however, there is no effect of radiation or air movement. Thus, the calculated WBGT would only be a function of temperature and humidity which are reported clearly and transparently in the manuscript (both in the psychrometric charts and in Table 3). We do recognize the importance, as highlighted by this reviewer, of better understanding how these indoor critical environmental limits would extend to outdoor settings that include solar radiation and wind. That is unfortunately outside the scope of the current study and, therefore, we feel that providing WBGT would currently be of limited value beyond the thermal indices provided by the psychrometric chart.

L 306-307: As this represents a rate and not total body water loss, it should be articulated as sweat loss per unit of body surface area per minute (e.g., mg/cm²/min). If this isn't the intended representation, I'd suggest rephrasing to "body water loss" throughout the manuscript.

We agree that this sentence was unclear. We have specified that sweat rate (in g·m⁻²·h⁻¹) was calculated after each experiment from the loss of nude body mass and that both percent body mass loss and sweat rate are reported.

Reviewer #2 (Remarks to the Author):

The manuscript titled “Heat stress vulnerability and critical environmental limits for older adults” evaluated older adults’ responses to a range of environmental conditions during seated rest and activity that represents typical daily living. The active responses were also compared to younger adults. These findings are the first to document age dependent vulnerability to adverse environmental conditions, and lower environmental limits in older compared to younger adults. The manuscript is clear, logical, and well written. The manuscript adopts robust methodological approaches that are leading within the domain of research, with large sample sizes and appropriate controls. The results are clear and well-presented, and the interpretation is logical and accurate. Overall, the manuscript provides novel insights into this domain of research and will act as an essential reference point for future research in this area. Due to the exceptional

production of this manuscript, I have little to comment on and would recommend this article to be accepted for publication.

Thank you.

Minor comments

The sample size (N = 51 younger; N = 49 older adults) is commendable and great to see so many females present within the sample size, however, I was left confused how your a priori power analysis led to the recruitment of such large sample sizes and whether this may have resulted in type 1 error.

We appreciate this concern from the reviewer. Because each subject completed just 2-4 experimental trials, a large sample was needed to be adequately powered in each environmental condition and at each metabolic rate (please see Figure 1).

Whilst the authors have acknowledged that no attempt was made to control for menstrual status or contraceptive use in young women, this is an experimental limitation, that should be addressed during the PSU HEAT projects long-term goal to determine sex differences in responses.

We recognize this limitation and have added it to the *Potential Limitations* section of the manuscript. The goal is to establish safe limits that are independent of menstrual cycle phase or contraceptive use, rather than attempting to establish separate thresholds for different phases of the cycle or for women who are/are not using contraceptives.